# Incorporating Object-Level Visual Context for Multimodal Fine-Grained Entity Typing

**Ying Zhang**[*], **Wenbo Fan, Kehui Song, Yu Zhao, Xuhui Sui, Xiaojie Yuan**
College of Computer Science, VCIP, TMCC, TBI Center, Nankai University, China
{yingzhang,yuanxj}@nankai.edu.cn
{fanwenbo,songkehui,zhaoyu,suixuhui}@dbis.nankai.edu.cn

## Abstract

Fine-grained entity typing (FGET) aims to assign appropriate fine-grained types to entity mentions within their context, which is an important foundational task in natural language processing. Previous approaches for FGET only utilized textual context information. However, in the form of short text, the contextual semantic information is often insufficient for FGET. In many real-world scenarios, text is often accompanied by images, and the visual context is valuable for FGET. To this end, we firstly propose a new task called multimodal fine-grained entity typing (MFGET). Then we construct a large-scale dataset for multimodal fine-grained entity typing called MFIGER based on FIGER. To fully leverage both textual and visual information, we propose a novel **M**ultimodal **O**bject-Level **V**isual **C**ontext **Net**work (**MOVCNet**). MOVCNet can capture fine-grained semantic information by detecting objects in images, and effectively merge both textual and visual context. Experimental results demonstrate that our approach achieves superior classification performance compared to previous approaches.

## 1 Introduction

Fine-Grained Entity Typing (FGET) aims to classify an entity mention with its context into one or more fine-grained types. For example, given a sentence "***Lionel Messi*** *won the championship of 2022 FIFA World Cup*", the mention "*Lionel Messi*" should be classified as *Person* as its coarse-grained type and *Athlete* as its fine-grained type. FGET serves many down-stream NLP applications, such as relation extraction (Liu et al., 2014) and entity linking (Onoe and Durrett, 2020; Sui et al., 2022), thus is the foundation for building knowledge graphs.

One major challenge of FGET lies in its rich and fine-grained labels with some kind of hierarchi-

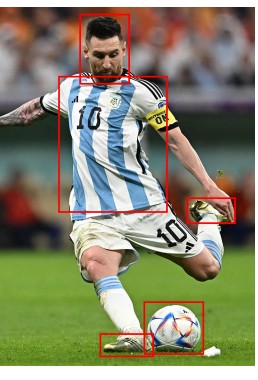
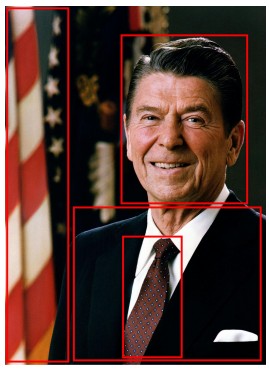

(a) **Lionel Messi [*Person, Athlete*]** won the championship of 2022 FIFA World Cup.

(b) **Ronald Reagan [*Person, Actor, Politician*]** auditioned for the movie The Philadelphia Story.

Figure 1: Two examples for Multimodal Fine-Grained Entity Typing (MFGET). Entity mentions and their fine-grained types in brackets are highlighted.

cal structure (Ling and Weld, 2012; Gillick et al., 2014; Choi et al., 2018). Without taking into account labels' interdependencies, it's hard to classify entity mention in isolation due to the large label set. Some recent works attempt to address this issue by leveraging label structures or statistics (Lin and Ji, 2019; Chen et al., 2020; Liu et al., 2021). Most of the previous methods only focus on textual content. However, in some certain situations, text can not provide sufficient contextual information to serve as the basis for classification, making it difficult to directly determine the ground truth label of the entity mention.

In many real-world scenarios, texts are often accompanied by images and the images also contain rich semantic information, which provides additional help to the FGET task. The advantages of incorporating visual contexts of images into textual contexts for FGET are summarized as follows:

1. **Visual context could enhance the indicative ability of textual context.** As shown in Figure 1(a), given the sentence "***Lionel Messi*** *won*

---

*the championship of 2022 FIFA World Cup*", we cannot accurately determine from the textual context whether the mention "*Lionel Messi*" refers to a person or a sports team. However, with the help of visual context, we can accurately classify the ambiguous "*Lionel Messi*" as *Person* and *Athlete* based on the detected objects in the image, e.g. sneakers and football.

2. **Visual context provides complementary semantics to the textual context.** As shown in Figure 1(b), given the sentence "***Ronald Reagan** auditioned for the movie The Philadelphia Story*", we can classify the mention "*Ronald Reagan*" as *Person* and *Actor* using the textual context "auditioned" and "movie". However, based on the visual context of objects like the American flag and suit, we could also infer its category as *Politician*. Together we infer all the ground truth labels of {*Person*, *Actor*, *Politician*}.

Therefore, considering that visual context is helpful for FGET, we try to introduce images into FGET, proposing a new task called Multimodal Fine-Grained Entity Typing (MFGET). In the meanwhile, we construct an MFGET dataset with a corresponding image for each sentence. The images are derived from Wikidata.[1] To incorporate visual information, we propose a multimodal object-level visual context network MOVCNet. MOVCNet can effectively extract local object features in the image that are relevant to the text. Through the object-based attention mechanism, MOVCNet can better fuse the text and image context and further improve the performance of fine-grained classification. Experimental results demonstrate the effectiveness of MOVCNet compared with previous approaches.

The main contributions of this work can be summarized as follows:

- To the best of our knowledge, we are the first to propose the task called Multimodal Fine-Grained Entity Typing (MFGET). Based on a widely used dataset FIGER for FGET, We construct a new dataset MFIGER for MFGET.

- We propose a multimodal object-level visual context network MOVCNet for MFGET. MOVCNet can effectively identify objects in images and fuse visual and textual context, aiding in classifying entity mention with its con-

---

[1] https://www.wikidata.org

text into fine-grained types.

- We evaluate the proposed method MOVCNet on our constructed dataset MFIGER. Compared with previous baselines, our method can effectively improve the performance of fine-grained entity typing.

## 2 Related Work

### 2.1 Fine-Grained Entity Typing

Two major challenges of FGET have been extensively studied by researchers. One challenge is that distant supervision introduces a significant amount of noise to FGET. Some researches divide the dataset into clean set and noisy set, and model them separately (Ren et al., 2016; Abhishek et al., 2017; Xu and Barbosa, 2018). Onoe and Durrett (2019) proposed a two-stage denoising method including filtering and relabeling function. Zhang et al. (2020) proposed a probabilistic automatic relabeling method with pseudo-truth label distribution estimation. Pan et al. (2022) proposed a method to correct and identify noisy labels. Pang et al. (2022) tried to mitigate the effect of noise by feature clustering and loss correction.

Another challenge in FGET is label hierarchy and interdependency. Xu and Barbosa (2018) introduced hierarchical loss normalization to deal with type hierarchy. Lin and Ji (2019) proposed a hybrid classification model to utilize type hierarchy. Chen et al. (2020) proposed a novel model with a multi-level learning-to-rank loss and a coarse-to-fine decoder. Liu et al. (2021) proposed a label reasoning network to capture extrinsic and intrinsic dependencies. Zuo et al. (2022) modeled type hierarchy by hierarchical contrastive strategy. Moreover, some works attempted to model FGET in hyperbolic space (López et al., 2019) or box space (Onoe et al., 2021) instead of traditional vector space.

However, images often co-occur with text in many real-world scenarios, yet no one has investigated the impact of images on FGET. Therefore, we introduce images to FGET and propose a new task called multimodal fine-grained entity typing, and then study the effect of images on FGET.

### 2.2 Multimodal Information Extraction

Multimodal information extraction aims to extract structured knowledge from various modalities, including unstructured and semi-structured text, images, videos, etc. There exists some tasks like multimodal named entity recognition(Wang et al.,

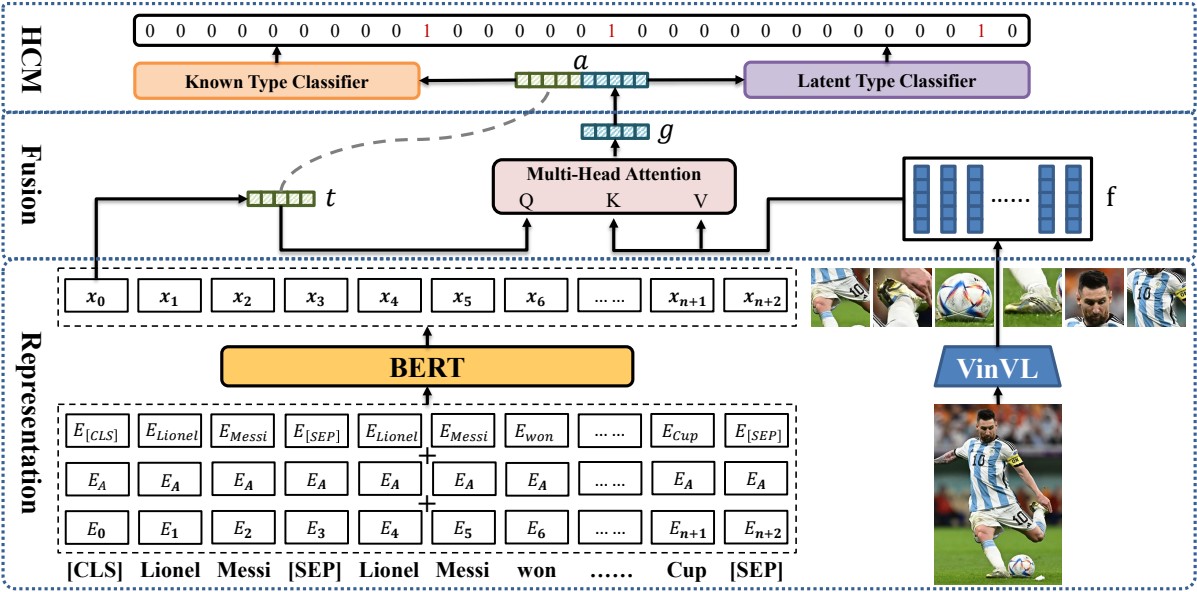

Figure 2: The overall architecture of our proposed model MOVCNet. "Representation" denotes the text and visual representation. "Fusion" denotes text-guided multimodal fusion. "HCM" denotes hybrid classification model.

2022b; Chen et al., 2022; Xu et al., 2022), multimodal relation extraction(Zhao et al., 2023; Chen et al., 2022), and multimodal entity linking(Zhang et al., 2023; Wang et al., 2022c). Information extraction techniques that incorporate multimodality form the foundation for constructing multimodal knowledge bases, providing ample data support for applications such as question-answering systems, information retrieval, and more.

Multimodal named entity recognition (MNER) aims to detect named entities and determine their corresponding entity types based on a {sentence, image} pair (Moon et al., 2018; Zhang et al., 2018; Lu et al., 2018). Multimodal relation extraction (MRE) aims to predict relations between two named entities in a sentence with the help of images (Zheng et al., 2021a,b). Multimodal entity linking (MEL) aims to map an ambiguous mention in a sentence to an entity in a knowledge base with textual and visual information (Adjali et al., 2020; Wang et al., 2022a). Most previous multimodal information extraction works focus on extracting better representations of both textual and visual modalities and designing better task-specific fusion models of two modalities.

However, existing multimodal information extraction methods cannot be directly applied to multimodal fine-grained entity typing. For example, existing MNER methods only consider coarse-grained labels like *Person*, *Location* and *Organization*, these coarse-grained labels can not provide a

precise characterization of the entities. Therefore, when applying them to MFGET, they cannot accurately classify entity mentions into their ground truth fine-grained labels. To this end, we propose a novel multimodal object-level visual context network MOVCNet for MFGET.

## 3 Methodolgy

In this section, we first introduce the definition of FGET and MFGET tasks. Next, we provide a detailed explanation of how to construct the MFIGER dataset. Then, we elaborate on the implementation details of our proposed model MOVCNet in four parts. Figure 2 shows the comprehensive architecture of MOVCNet.

### 3.1 Definition

Traditional fine-grained entity typing datasets consist of a massive collection of (mention, context) tuples: $\mathcal{D} = \{(m_1, c_1),(m_2, c_2),...,(m_n, c_n)\}$. Given an instance $(m, c)$, the FGET task aims to predict its appropriate types $y \subseteq \mathcal{T}$, where $\mathcal{T} = \{t_1, t_2, ..., t_{|\mathcal{T}|}\}$ is the pre-defined fine-grained type set and $|\mathcal{T}|$ is the number of candidate types.

Different from FGET, multimodal fine-grained entity typing datasets consist of a massive collection of (mention, context, image) tuples: $\mathcal{D} = \{(m_1, c_1, v_1),(m_2, c_2, v_2),...,(m_n, c_n, v_n)\}$. The additional image information helps models predict corresponding types more accurately.

## 3.2 Dataset Construction

FIGER was proposed by (Ling and Weld, 2012), which contains 113 types with a 2-level hierarchy. Based on FIGER, we construct a new dataset **MFIGER** for Multimodal Fine-Grained Entity Typing. The detailed procedure consists of the following four steps:

1. Based on a mapping file from Wikipedia titles to Freebase mids, we first retrieve the Freebase mid of each mention within its context.

2. Based on a mapping file from Freebase entities to Wikidata entities, we get the Wikidata id of each entity mention within its context.

3. We get images from the Wikidata webpage according to the Wikidata id, ultimately obtaining one image corresponding to each entity mention within its context.

4. The original size of FIGER is 2,010,563, and after the above three steps, 935,744 instances have images. We divide them into the training set, validation set, and test set by 7:1:2.

MFIGER contains 102 types with a 2-level hierarchy like FIGER. And the statistical information is presented in Section 4.1.

## 3.3 Text Representation

Following (Onoe et al., 2021; Ding et al., 2022; Pan et al., 2022), we adopt BERT (Devlin et al., 2019) as our text encoder, the input of BERT is the sentence represented as $S$ = [CLS] mention [SEP] context [SEP]. The output is $X = [x_0, x_1, ..., x_{n+2}]$ including 3 special tokens, where $x_i \in \mathbb{R}^{d_j}$ denotes the contextualized representation of the $i$-th word in the sentence, $d_j$ denotes the dimension of hidden layer in BERT, and $n$ denotes the length of the input sentence. Finally, by taking the hidden vector at [CLS] token, we encode the whole sequence into a single vector $t$:

$$t = \text{BERT}(S; \theta^{bert}) \in \mathbb{R}^{d_j} \qquad (1)$$

where $\theta^{bert}$ is the parameter of BERT encoder, $d_j = 768$ is the dimension of BERT hidden state.

## 3.4 Object-Level Visual Representation

For the visual representation, traditional CNNs like VGG (Simonyan and Zisserman, 2015) and ResNet (He et al., 2016) can effectively extract global features of an image. However, the types in MFGET are fine-grained with some kind of hierarchy, so it is crucial to extract highly informative and nuanced features from images. The objects in an image can be used to deduce the fine-grained type of an entity mention, e.g. a badminton racket implies that he is an athlete. So by extracting local object features in an image, we can improve the precision and effectiveness of MFGET.

Thus, we adopt the object detector of VinVL (Zhang et al., 2021) as our visual encoder, which is a large pre-trained vision-language model and contains a large-scale object-attribute detection model based on ResNeXt-152 C4 architecture. Given an image $v$, we extract top $m$ local visual objects as follows:

$$f = \text{VinVL}(v) \in \mathbb{R}^{m \times d_v} \qquad (2)$$

where $m$ is the number of objects, $d_v = 2048$ is the dimension of object feature representation.

## 3.5 Text-Guided Multimodal Fusion

We use Multi-Head Attention (Vaswani et al., 2017) to effectively fuse textual and visual context information. To align the dimensions of the both textual and visual representation, we add a fully connected layer on the visual representation $f$ as follows:

$$p = W^f f \in \mathbb{R}^{m \times d_j} \qquad (3)$$

where $W^f \in \mathbb{R}^{d_j \times d_v}$ is a trainable parameter.

Specifically, we treat the textual representation $t$ as the query, and the transformed visual representation $p$ as the key and value. We get text-aware visual representation $g$ as follows:

$$\begin{aligned} g &= \text{MultiHead}(Q, K, V) \\ &= \text{Concat}(head_1, ..., head_h)W^O \end{aligned} \qquad (4)$$

where $h$ is the number of attention head, $head_i$ represents the output of the $i$-th attention head, and $W^O$ is the output transformation matrix. The output of each head $head_i$ can be calculated as follows:

$$head_i = \text{Attention}(Q_i, K_i, V_i), i = 1, ..., h \qquad (5)$$

where Attention is the calculation function of attention, and

$$\begin{aligned} Q_i &= QW_i^Q, \\ K_i &= KW_i^K, \\ V_i &= VW_i^V, i = 1, ..., h \end{aligned} \qquad (6)$$

where $\boldsymbol{W}_i^Q$, $\boldsymbol{W}_i^K$, $\boldsymbol{W}_i^V$ are the transformation matrix of the $i$-th query, key and value respectively.

We get the final representation $\boldsymbol{a}$, which is the concatenation of textual representation $\boldsymbol{t}$ and text-aware visual object representation $\boldsymbol{g}$.

$$\boldsymbol{a} = \text{Concat}(\boldsymbol{t}, \boldsymbol{g}) \qquad (7)$$

### 3.6 Hybrid Classification Model

Following Lin and Ji (2019), we use a hybrid type classification model consisting of two classifiers: known type classifier and latent type classifier.

Our known type classifier trains a linear transformation matrix $\boldsymbol{W}^a$ to independently predict each type without considering their interdependencies:

$$\tilde{\boldsymbol{y}}^a = \boldsymbol{W}^a \boldsymbol{a} \qquad (8)$$

where $\tilde{\boldsymbol{y}}_i^a$ is the predicted probability for the $i$-th type, $\boldsymbol{W}^a \in \mathbb{R}^{d_n \times 2d_j}$ and $d_n$ is the number of types.

To fully leverage the type interdependency and hierarchy, we use a latent type classifier motivated by Principle Label Space Transformation (Tai and Lin, 2012). Based on the hypercube sparsity assumption, where $2^{d_n}$ is significantly larger than the size of the training set, Tai and Lin (2012) utilize Singular Value Decomposition (SVD) to reduce the dimensionality of high-dimensional type vectors by projecting them into a lower-dimensional space. This projection allows us to uncover the underlying type correlations that go beyond first-order co-occurrence. The formula of SVD is as follows:

$$\boldsymbol{Y} \approx \tilde{\boldsymbol{Y}} = \boldsymbol{U}\boldsymbol{\Sigma}\boldsymbol{L}^\top \qquad (9)$$

where $\boldsymbol{U} \in \mathbb{R}^{d_n \times d_l}$, $\boldsymbol{\Sigma} \in \mathbb{R}^{d_l \times d_l}$, $\boldsymbol{L} \in \mathbb{R}^{N \times d_l}$, and $d_n \gg d_l$. The resulting low-dimensional space resembles the hidden concept space found in Latent Semantic Analysis (Deerwester et al., 1990). Each row of the matrix $\boldsymbol{L}$ represents the latent representation of a specific type vector. Subsequently, we predict the latent type representation from the feature vector:

$$\boldsymbol{l} = \boldsymbol{V}^l \boldsymbol{a} \qquad (10)$$

where $\boldsymbol{V}^l \in \mathbb{R}^{d_l \times 2d_j}$ is a trainable parameter. Using a linear transformation matrix $\boldsymbol{W}^l$, we reconstruct the type vector based on $\boldsymbol{l}$:

$$\tilde{\boldsymbol{y}}^b = \boldsymbol{W}^l \boldsymbol{l} = \boldsymbol{U}\boldsymbol{\Sigma}\boldsymbol{l} \qquad (11)$$

where $\tilde{\boldsymbol{y}}_i^b$ is the predicted probability for the $i$-th type, $\boldsymbol{W}^l \in \mathbb{R}^{d_n \times d_l}$ is a trainable parameter.

| No. | Coarse | #Fine | Train | Dev | Test |
|------|--------------|-------|---------|--------|---------|
| C1 | Person | 14 | 217,430 | 30,971 | 61,992 |
| C2 | Location | 13 | 314,283 | 44,948 | 90,045 |
| C3 | Organization | 12 | 118,016 | 16,820 | 33,607 |
| C4 | Art | 4 | 19,444 | 2,727 | 5,414 |
| C5 | Event | 6 | 27,140 | 3,899 | 7,778 |
| C6 | Building | 6 | 27,370 | 3,889 | 7,802 |
| C7 | Product | 10 | 12,904 | 1,885 | 3,634 |
| C8 | Others | 30 | 62,320 | 8,744 | 17,817 |
| Total | 8 | 95 | 655,022 | 93,574 | 187,148 |

Table 1: MFIGER type statistics.[2]

Finally, we combine the above two classifiers using the following formula:

$$\tilde{\boldsymbol{y}} = \tilde{\boldsymbol{y}}^a + \lambda \tilde{\boldsymbol{y}}^b \qquad (12)$$

where $\tilde{\boldsymbol{y}}_i$ is the overall predicted probability for the $i$-th type of the two classifiers. $\lambda$ is a scalar with an initial value of 0.1, and $\lambda$ is dynamically adjusted during the training phase.

We regard MFGET as a multi-label classification problem, so a multi-label training objective is needed. We optimize a multi-label binary cross-entropy-based objective:

$$\mathcal{L} = -\frac{1}{N}\sum_i^N \boldsymbol{y_i} \log \tilde{\boldsymbol{y}_i} + (1 - \boldsymbol{y_i}) \log(1 - \tilde{\boldsymbol{y}_i}) \qquad (13)$$

where $y_i$ is set the value 1 if the mention is classified as the $i$-th type, $N$ is the number of types.

During the test phase, we make predictions for each type based on the probability $\tilde{y}_i > 0.5$. If all probabilities are lower than 0.5, we select the type with the highest probability using $\arg\max \tilde{y}_i$.

## 4 Experiments

In this section, we compare our proposed method with previous state-of-the-art approaches to validate the effectiveness of our model. We first introduce the dataset and the statistical information that we use. Then, we provide a brief overview of the baseline models for comparison and our implementation details. Finally, we present the overall results and analysis on the performance comparison between our model and others.

### 4.1 Datasets

We evaluate our model on our own constructed multimodal dataset MFIGER, which comprises pairs of sentences with annotated entity mentions and their associated images. The detailed explanation

---

[2]Note that an entity mention may have multiple types.

| Model | Total | | | Coarse | | | Fine | | |
|---|---|---|---|---|---|---|---|---|---|
| | Acc | Ma-F1 | Mi-F1 | Acc | Ma-F1 | Mi-F1 | Acc | Ma-F1 | Mi-F1 |
| **NFETC** (2018) | 54.23 | 85.53 | 79.21 | 82.04 | 93.25 | 89.26 | 57.12 | 78.69 | 69.78 |
| Lin and Ji (2019) | 72.80 | 92.13 | 91.58 | 93.59 | 96.97 | 96.23 | 75.45 | 88.49 | 87.91 |
| **ML-L2R** (2020) | 47.99 | 84.19 | 79.44 | 84.43 | 93.69 | 90.88 | 52.43 | 75.06 | 68.20 |
| **Box** (2021) | 79.72 | 94.20 | 93.67 | 95.63 | 97.88 | 97.39 | 81.21 | 91.27 | 90.72 |
| **NFETC-FCLC** (2022) | 48.93 | 83.15 | 76.32 | 82.05 | 93.34 | 89.32 | 53.10 | 73.07 | 63.29 |
| **DenoiseFET** (2022) | 67.83 | 90.87 | 89.72 | 93.05 | 96.68 | 95.87 | 69.91 | 86.48 | 85.03 |
| **UMT** (2020) | 88.56 | 95.93 | 96.09 | 96.95 | 98.29 | 98.05 | 89.43 | 94.05 | 94.56 |
| **MAF** (2022) | 77.17 | 92.07 | 92.04 | 94.11 | 96.70 | 96.22 | 78.69 | 88.67 | 88.69 |
| **MOVCNet** | **90.99** | **96.80** | **96.92** | **97.54** | **98.65** | **98.44** | **91.67** | **95.27** | **95.70** |

Table 2: Overall results of different label granularity on MFIGER test set. "Coarse" represents the results on 8 coarse-grained types, "Fine" represents the results on fine-grained types. The **best** results are highlighted.

of the dataset construction process is in Section 3.2. We have summarized 8 coarse-grained categories and 95 fine-grained categories. Table 1 presents the type statistics of the dataset. We can see that train, dev, and test set share a similar type distribution.

## 4.2 Baselines

For multimodal fine-grained entity typing task, we compare our model with the following text-based and multimodal state-of-the-art approaches:

- **NFETC** (Xu and Barbosa, 2018) tries to solve noisy label problems by a variant of the cross-entropy loss function, and deals with type hierarchy by hierarchical loss normalization.

- **Lin and Ji** (Lin and Ji, 2019) presents a two-step attention mechanism and a hybrid classification method to utilize label co-occurrence.

- **ML-L2R** (Chen et al., 2020) proposes a novel multi-level learning-to-rank especially for hierarchical classification problems.

- **Box** (Onoe et al., 2021) is the first to introduce box space to FGET instead of traditional vector space.

- **NFETC-FCLC** (Pang et al., 2022) designs a feature-clustering method with loss correction on each cluster.

- **DenoiseFET** (Pan et al., 2022) proposes a method to automatically correct and identify noisy labels.

- **UMT** (Yu et al., 2020) designs a unified multimodal transformer with an entity span detection module which can better capture the intrinsic correlations between modalities.

- **MAF** (Xu et al., 2022) proposes a matching and alignment framework to make text and image more consistent.

## 4.3 Implementation Details

We adopt BERT-Base(cased) (Devlin et al., 2019) as encoder, Adam optimizer (Kingma and Ba, 2015) with a learning rate of BERT at 5e-5. The training batch size is 32, the hidden size of BERT encoder is 768, and the dropout rate is 0.1. For VinVL (Zhang et al., 2021), we set the number of local visual objects $m = 10$. For Multi-Head Attention, we set the number of attention head $h = 4$. Our experiments are conducted on *NVIDIA RTX 2080 Ti* GPUs, and all models are implemented using PyTorch. Our experimental code is available here [2].

Following previous work (Ling and Weld, 2012), we use strict accuracy(Acc), macro-averaged F1 score(Ma-F1), and micro-averaged F1 score(Mi-F1) to evaluate the performance of models.

## 4.4 Overall Results

Table 2 shows the overall results of different label granularity of all baselines and our method on our own constructed dataset MFIGER test set. We can clearly see that our proposed method MOVCNet significantly outperforms previous methods, whether at the total level, at the coarse-grained level, or at the fine-grained level.

At the total level, we get the evaluation results of all baselines and our method on total of 102 labels, including 8 coarse-grained labels and 95 fine-grained labels. Compared with previous SOTA method UMT (Yu et al., 2020), MOVCNet achieves 2.43% improvement in strict accuracy (from 88.56% to 90.99%), 0.87% improvement on

[2] https://github.com/Web-FAN/MOVCNet

| Model | Total | | | Coarse | | | Fine | | |
|---|---|---|---|---|---|---|---|---|---|
| | Acc | Ma-F1 | Mi-F1 | Acc | Ma-F1 | Mi-F1 | Acc | Ma-F1 | Mi-F1 |
| **MOVCNet** | **90.99** | **96.80** | **96.92** | **97.54** | **98.65** | **98.44** | **91.67** | **95.27** | **95.70** |
| w/o BERT | 87.60 | 95.71 | 95.91 | 96.70 | 98.19 | 97.94 | 88.69 | 93.78 | 94.34 |
| w/o object | 89.98 | 96.06 | 96.29 | 97.11 | 98.27 | 98.11 | 90.62 | 94.33 | 94.93 |
| w/o attention | 90.75 | 96.37 | 96.60 | 97.23 | 98.39 | 98.21 | 91.40 | 94.84 | 95.38 |

Table 3: Ablation study of different label granularity on MFIGER. "w/o BERT" denotes replacing BERT with ELMo. "w/o object" denotes replacing object features with global image features from VGG16. "w/o attention" denotes replacing multi-head attention with average pooling.

macro-averaged F1 score (from 95.93% to 96.80%), 0.83% improvement on micro-averaged F1 score (from 96.09% to 96.92%).

At the coarse-grained level, we get the evaluation results of all baselines and our method only on 8 coarse-grained labels. Compared with previous SOTA method UMT (Yu et al., 2020), MOVCNet achieves 0.59% improvement on strict accuracy (from 96.95% to 97.54%), 0.36% improvement on macro-averaged F1 score (from 98.29% to 98.65%), 0.39% improvement on micro-averaged F1 score (from 98.05% to 98.44%). Besides, most baseline models perform well on coarse-grained entity typing, achieving an accuracy of over 80%, and some models even surpass 90%. This indicates that the coarse-grained entity typing task is relatively simple, as the coarse-grained types of entity mentions can be accurately inferred from the contextual information contained in the text alone.

At the fine-grained level, we get the evaluation results of all baselines and our method only on 95 fine-grained labels. Compared with the previous SOTA method UMT (Yu et al., 2020), MOVCNet achieves 2.24% improvement in strict accuracy (from 89.43% to 91.67%), 1.22% improvement on macro-averaged F1 score (from 94.05% to 95.27%), 1.14% improvement on micro-averaged F1 score (from 94.56% to 95.70%).

Compared to the coarse-grained level, our multimodal model MOVCNet shows a greater improvement at the fine-grained level. The fine-grained entity typing task is relatively complex, as it is difficult to accurately infer the fine-grained category of an entity mention based solely on the textual context or a combination of textual context and global image features. Our multimodal model introduces visual context from images, effectively leveraging the objects contained in the images and interacting with the textual context for fusion. Through this approach, the images can effectively assist in classifying the entity mentions into fine-grained

categories, ultimately improving the performance of fine-grained entity typing, including strict accuracy, micro-averaged F1 score, and micro-averaged F1 score.

## 4.5 Ablation Study

To study the effects of different modules in our model, we design three variants of MOVCNet. Table 3 shows the results of the ablation study of different label granularity on our dataset MFIGER.

**Effect of BERT.** We replace BERT with ELMo (Peters et al., 2018) as our sentence encoder, as (Lin and Ji, 2019) is the first to use ELMo to get contextualized word representations. Compared with ELMo, BERT brings 3.39% improvement on strict accuracy, 1.09% improvement on macro-averaged F1 score. This demonstrates that in the MFGET task, BERT is more capable of obtaining better text representations, which in turn facilitates the fusion with image representations.

**Effect of Object Feature.** We replace the local object features detected by VinVL with global image features extracted by traditional Convolutional Neural Network (CNN) VGG16 (Simonyan and Zisserman, 2015). Compared with VGG16, the object feature brings 1.01% improvement in strict accuracy, 0.74% improvement in macro-averaged F1 score. VGG can extract global features from images, but it may not capture certain details. On the other hand, VinVL can effectively extract objects in the image, and these objects provide fine-grained visual context to aid in fine-grained entity typing.

**Effect of Attention Mechanism.** We replace Multi-Head Attention with the average representation over top $m$ objects. Compared with average representation, Multi-Head Attention brings 0.24% improvement on strict accuracy, 0.43% improvement in macro-averaged F1 score. With the attention mechanism, we can determine which objects are relevant to the entity mentions in the sentence, and thereby obtain image representations that are

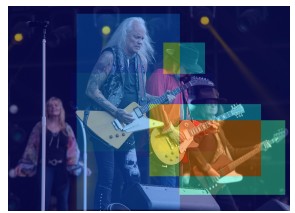 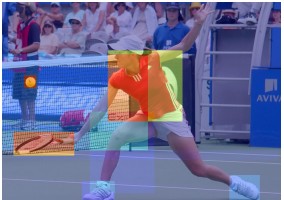

(a) It is a song by southern rock band **Lynyrd Skynyrd** [*Person, Musician*] released on its 1974 album.

(b) **Justine Henin** [*Person, Athlete*] won her first tournament since her comeback at the Tennis Grand.

Figure 3: Two examples of attention visualization. The more the color tends towards red, the higher the weight of attention.

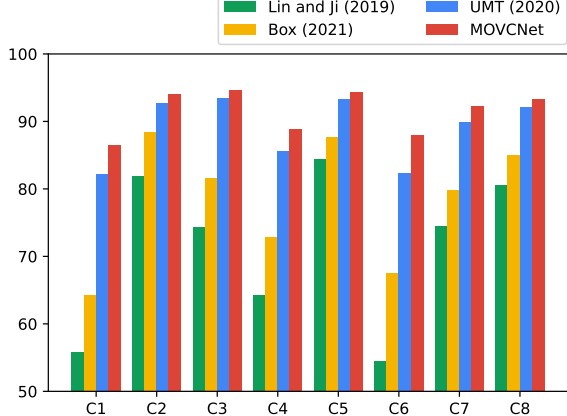

Figure 4: Results across 8 coarse-grained types respectively. C1-C8 refers to the 8 coarse-grained types described in Table 1. Y-axis refers to the strict classification accuracy (%), it begins from 50%.

perceptually aligned with the text. This allows the most relevant objects to assist in fine-grained entity typing, improving the classification performance.

Based on the analysis of the above three variants, we can draw the following conclusions. In our model, the module that has the greatest positive impact on multimodal fine-grained entity typing is BERT, followed by the detected objects, and finally the attention mechanism.

### 4.6 Attention Visualization

To study the effectiveness of the attention mechanism in our model MOVCNet, we visualized the attention in Figure 3. In figure 3(a), the ground truth type is *Person* and *Musician*. We can see that the guitars are given more attention, providing strong visual cues for the classification of *Musician*. In figure 3(b), the ground truth type is *Person* and *Athlete*. We can see that the tennis ball, the tennis racket, and the sportswear are given more attention, providing useful visual context for the classification of *Athlete*. These two examples demonstrate that the Multi-Head Attention used in our model can effectively extract visual contextual information from the images, and the visual cues are relevant to the entity mention in the sentence. Therefore, MOVCNet can achieve better classification performance than previous models.

### 4.7 Results on Different Categories

To further analyze the effect of images in fine-grained entity typing task, we conduct a comparative experiment on the classification performance of our model and three baseline models. Figure 4 shows the fine-grained classification results summarized across 8 coarse-grained types respectively, there are several fine-grained types under each coarse-grained type.

Compared with previous baselines, our model achieves significant improvements in the strict classification accuracy on 8 coarse-grained types. The gain in classification performance, from highest to lowest, is as follows: *Person* (C1), *Building* (C6), *Art* (C4), *Organization* (C3), *Product* (C7), *Others* (C8), *Location* (C2), *Event* (C5).

We can infer that the fine-grained types under the coarse-grained type *Person* are relatively complex, such as *Actor*, *Musician* and *Politician*, as it requires strong textual cues to indicate a person's profession. Given an entity mention within a sentence with *Person* as its ground truth label, we can hardly classify it into specific fine-grained types directly. Because the textual context alone is always limited, it may not contain sufficient fine-grained information to assist in fine-grained classification. Besides, the global features of an image do not contain sufficient fine-grained object information. Similarly, for mentions belonging to *Building* or *Art*, it is also hard to determine their fine-grained types from the textual context or global image features.

Under the circumstances, our model introduces object-level image information to FGET. MOVCNet can extract the most relevant local objects in images and integrate information from both modalities effectively. So MOVCNet can provide important visual cues to the textual context, e.g. guitar or racket for *Person*, parking apron for *Building*, piano sheet music for *Art*. These objects are valuable for identifying the profession of a person, the type of a

building or an art, so the classification performance of our model can significantly surpass that of the compared baseline models across 8 coarse-grained types respectively.

## 5 Conclusion

In this paper, we propose a new task called multi-modal fine-grained entity typing (MFGET). Based on FIGER, we construct a multimodal dataset MFIGER with both text and accompanying images for MFGET. We propose a novel multimodal model MOVCNet to incorporate object-level visual context for FGET. Specifically, MOVCNet can capture relevant objects in images and merge visual and textual contexts effectively. Experimental results on MFIGER demonstrate that our proposed model achieves the best performance compared with competitive existing models.

## Limitations

Although our model MOVCNet has achieved excellent results, it should be noted that we have used a simple off-the-shelf object detection tool VinVL that can effectively extract the objects from the images and get their features. However, there may be better methods for object detection, or we can design a dedicated object detection method specifically for multimodal fine-grained entity typing to better extract local object features that are relevant to the text. These areas can be further explored in future work.

## Acknowledgements

This research is supported by the National Natural Science Foundation of China (No. 62272250), the Natural Science Foundation of Tianjin, China (No. 22JCJQJC00150, 22JCQNJC01580), the Fundamental Research Funds for the Central Universities (No. 63231149), Tianjin Research Innovation Project for Postgraduate Students (No. 2022SKYZ232).

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
