# OpenReview forum: "Incorporating Object-Level Visual Context for Multimodal Fine-Grained Entity Typing"
_EMNLP/2023/Conference — EMNLP 2023 Findings_

### Official Review · Reviewer_JYT1 · 2023-07-31

**Soundness:** 2

**Excitement:**

2: Mediocre: This paper makes marginal contributions (vs non-contemporaneous work), so I would rather not see it in the conference.

**Missing References:**

1. Chen, Xiang, Ningyu Zhang, Lei Li, Yunzhi Yao, Shumin Deng, Chuanqi Tan, Fei Huang, Luo Si, and Huajun Chen. "Good visual guidance makes a better extractor: Hierarchical visual prefix for multimodal entity and relation extraction."NAACL 2022.
2. Bo Xu, Shizhou Huang, Chaofeng Sha, and Hongya Wang. 2022. MAF: A General Matching and Alignment Framework for Multimodal Named Entity Recognition. In Proceedings of the Fifteenth ACM International Conference on Web Search and Data Mining (WSDM '22). Association for Computing Machinery, New York, NY, USA, 1215–1223. https://doi.org/10.1145/3488560.3498475
3. C. Zheng, Z. Wu, T. Wang, Y. Cai and Q. Li, "Object-Aware Multimodal Named Entity Recognition in Social Media Posts With Adversarial Learning," in IEEE Transactions on Multimedia, vol. 23, pp. 2520-2532, 2021, doi: 10.1109/TMM.2020.3013398.
4. Zhiwei Wu, Changmeng Zheng, Yi Cai, Junying Chen, Ho-fung Leung, and Qing Li. 2020. Multimodal Representation with Embedded Visual Guiding Objects for Named Entity Recognition in Social Media Posts. In Proceedings of the 28th ACM International Conference on Multimedia (MM '20). Association for Computing Machinery, New York, NY, USA, 1038–1046. https://doi.org/10.1145/3394171.3413650


**Paper Topic And Main Contributions:**

This paper first construct a new dataset for multimodal fine-grained entity typing based on FIGER. They propose a multimodal object-level visual context network, which fuses textual information and object-level visual information through attention layers, to capture fine-grained semantic information. Experiments compared with several existing textual-level models show that fusing object-level visual information can help improve the performance.

**Questions For The Authors:**

A. For section 3.2 dataset construction. What's the quality of the collected images? Is there any method to evaluate your images quality? Does there exist any noise for the images?
B. Line 302, is the attention calculation following self-attention mechanism, where q is from text and k and v are from images?
C. Line 356-357, could you please explain more about how you dynamically adjust the parameter \lambda? what's the final value for \lambda?
D. For your experiments, all baselines are unimodal. In other words, they only use textual information. There are lots of existing multimodal fusion models, I would like to see how your model compare with those multimodal fusion models. You could refer to the missing references, or even some simpliest fusion techniques such as concatenation, circulant fusion, etc. If there is no such comparison, the paper only proves that adding object-level visual information works better than only using textual information. In general, if the assumption is you have high quality inputs (additional information such as images), it always help improve the performance (regardless of efficiency). In this way, it is hard to get the contribution of your proposed multimodal object-level visual context network. Because in your ablation study, using your attention-based fusion method, the performance improves less than 1 percentage. The most performance contribution is provided by adding additional information for the inputs.


**Reasons To Accept:**

A new dataset is constructed for multimodal fine-grained entity typing task and their proposed model shows that adding object-level visual information can improve the performance a lot. The attention visualization helps better understand how object-level visual information works.

**Reasons To Reject:**

The main concerns of this paper are:
1. Fusing object information by cross-modal attention layeres are commonly used for multi-modal information extraction. Please refer to missing references section. And the hybrid classification layer is also following previous works.
2. The experiment is only conducted on one dataset. Need to implement more multi-modal baselines for performance comparison. Please see details in question D.

**Reproducibility:**

2: Would be hard pressed to reproduce the results. The contribution depends on data that are simply not available outside the author's institution or consortium; not enough details are provided.

**Reviewer Confidence:**

4: Quite sure. I tried to check the important points carefully. It's unlikely, though conceivable, that I missed something that should affect my ratings.

---

> ### Author Rebuttal · Authors · 2023-08-29
>
> We thank the reviewer for the insightful comments and sincerely appreciate your feedback. We would like to address the specific questions below.
>
> 1.Commonly used object information fusion and the hybrid classification layer
>
> (1) For multi-modal fusion, although fusing object information by cross-modal attention layers is commonly used for multi-modal information extraction, no one has introduced object information into fine-grained entity typing task. Besides, entity mentions are required to be classified into many fine-grained types, detected objects in images will provide strong visual cues for fine-grained classification. (2)For hybrid classification layer, Lin and Ji (2019) exploits type interdependency with latent type representation and improves classification performance significantly. So we adopt the latent type classifier following Lin and Ji (2019). However, our main contribution is to introduce visual information into fine-grained entity typing and propose a new task. So we are more focused on the effect of visual information and the fusion of two modalities, rather than the hybrid type classifier.
>
> 2.Experiment on only one dataset
>
> We are the first to propose the new task, and the dataset construction process is complex. So we only conduct experiments on our own constructed dataset.
>
> 3.Dataset quality and noise
>
> We get images from Wikidata webpages, so the image quality is relatively high, while inevitably there will be some noise. We can evaluate image quality by calculating the similarity between images and texts.
>
> 4.Attention calculation
>
> Yes, we use multi-head attention, and q is from text and k and v are from images.
>
> 5.Parameter lambda
>
> Lambda is a nn.Parameter(), it's fed into the optimizer and updates with training. The final value is 0.1474.
>
> 6.Missing multi-modal baselines for performance comparison
>
> Our main contribution is to introduce images into fine-grained entity typing. As shown in Table 2 and Table 3, the classification performance of our model is better than others, this indicates the effectiveness of our model. As shown in Table 3 in “4.5 Ablation Study”, “w/o object” denotes concatenating textual and VGG16 visual representation, “w/o attention” denotes concatenating textual and average pooling object visual representation, they are both simple concatenation. Besides, we have implemented some representative multi-modal baselines like UMT and MAF, the performance of our model is better than them. This further validates the effectiveness of our model. We will include this part of comparison experiment in the revised version.
>
> | Model | Total_Acc | Total_Ma-F1 | Total_Mi-F1 | Coarse_Acc | Coarse_Ma-F1 | Coarse_Mi-F1 | Fine_Acc | Fine_Ma-F1 | Fine_Mi-F1 |
>
> | UMT | 88.56    |     95.93        |      96.09      |      96.95       |        98.29         |          98.05      |    89.43    |       94.05     |       94.56    |
>
> | MAF   |    77.17    |     92.07        |      92.04      |      94.11       |        96.70         |          96.22      |    78.69    |       88.67     |       88.69    |
>
> | MOVCNet  |    90.99    |     96.80        |      96.92      |      97.54       |        98.65         |          98.44      |    91.67    |       95.27     |       95.70    |

---

### Official Review · Reviewer_mhGN · 2023-08-04

**Typos Grammar Style And Presentation Improvements:** N/A
**Soundness:** 5

**Excitement:**

4: Strong: This paper deepens the understanding of some phenomenon or lowers the barriers to an existing research direction.

**Missing References:**

N/A

**Paper Topic And Main Contributions:**

- The first work to present this multimodal fine-grained entity typing which predicts entity types with the aid of visual context
- Curate MFIGER, dataset on this task with 95 fine-grained categories
- Purpose MOVCNet that leverages object-level attention mechanism
- Strong performance gain on MFIGER compared to baselines

**Questions For The Authors:**

Question A: Recent years there have been a few multimodal models such as OpenFlamingo [1] or LLaVA [2]. I wonder what might be the zero shot performance of those model on coarse level, e.g. simply asking the model which one of the label does the entity belong to. However I understand that running those models might take a lot of compute so my score will not be affected by the results.

[1]: https://github.com/mlfoundations/open_flamingo

[2]: https://arxiv.org/abs/2304.08485

**Reasons To Accept:**

- The first to purpose multimodal fine-grained entity typing task
- 11.27%, 1.91%, and 10.46% strict accuracy improvement over baselines on total level, coarse level and fine-grained level on MFIGER
- MFIGER datasets can be a useful benchmark for future research and is beneficial for the community.
- Motivation is clear and I like the careful model design

**Reasons To Reject:**

No such thing, I believe this is a sound paper.

**Reproducibility:**

4: Could mostly reproduce the results, but there may be some variation because of sample variance or minor variations in their interpretation of the protocol or method.

**Reviewer Confidence:**

3: Pretty sure, but there's a chance I missed something. Although I have a good feel for this area in general, I did not carefully check the paper's details, e.g., the math, experimental design, or novelty.

---

> ### Author Rebuttal · Authors · 2023-08-29
>
> We thank the reviewer for the constructive suggestions and sincerely appreciate your positive feedback. We would like to address the specific questions below.
>
> 1.Zero-shot performance of the multimodal models on coarse level
>
> Large multimodal models require a lot of compute, and they focus more on unsupervised and zero-shot scenarios. Our model focuses on supervised scenarios, and will get better performance than those multimodal models.

---

### Official Review · Reviewer_MZdU · 2023-08-07

**Soundness:** 3

**Excitement:**

3: Ambivalent: It has merits (e.g., it reports state-of-the-art results, the idea is nice), but there are key weaknesses (e.g., it describes incremental work), and it can significantly benefit from another round of revision. However, I won't object to accepting it if my co-reviewers champion it.

**Paper Topic And Main Contributions:**

- This paper propose a novel task called Multimodal Fine-Grained Entity Typing (MFGET), and construct a new dataset MFIGER for MFGET task.
- This paper propose a novel MFGET method called MOVCNet, which can identify objects in images and fuse visual and textual context, aiding in classifying entity mention with its context into fine-grained types.
- Compared with previous text-only baselines, MOVCNet achieves better performances in the proposed MFIGER dataset.

**Reasons To Accept:**

- This paper is well-written and is easy to follow.
- The attention visualization enhances the interpretability of the proposed model.

**Reasons To Reject:**

- Missing ablation study on the latent type classifier. Does the latent type classifier help to improve the classification accuracy?
- Missing some comparison experiments of baselines. For example, for other text-only methods, can VinVL features be introduced into text features through attention? I think such a comparison is more fair.

**Reproducibility:**

4: Could mostly reproduce the results, but there may be some variation because of sample variance or minor variations in their interpretation of the protocol or method.

**Reviewer Confidence:**

3: Pretty sure, but there's a chance I missed something. Although I have a good feel for this area in general, I did not carefully check the paper's details, e.g., the math, experimental design, or novelty.

---

> ### Author Rebuttal · Authors · 2023-08-29
>
> We thank the reviewer for the insightful comments and sincerely appreciate your feedback. We would like to address the specific questions below.
>
> 1.Missing ablation study on the latent type classifier
>
> Lin and Ji (2019) exploits type interdependency with latent type representation and improves classification performance significantly, so we adopt the latent type classifier following Lin and Ji (2019). Our main contribution is to introduce visual information into fine-grained entity typing and the latent type classifier is not the technical innovation of our model. Therefore, in the ablation study, we are more focused on our contribution including object information and object-level attention, rather than the latent type classifier.
>
> 2.Missing comparison experiments of baselines
>
> As shown in Table 3, "w/o BERT" denotes introducing VinVL features into Lin and Ji (2019). The performance of "w/o BERT" is lower than our model MOVCNet, but higher than Lin and Ji (2019).

---

### Meta-Review · Area_Chair_tBdt · 2023-09-19

**Recommendation:** 2

**Metareview:**

Summary:
This paper begins by creating a new dataset for multimodal fine-grained entity typing, building upon FIGER. The authors introduce a multimodal object-level visual context network, which employs attention layers to fuse textual information and object-level visual data, facilitating the capture of fine-grained semantic information. Comparative experiments with several existing textual-level models demonstrate that the integration of object-level visual information enhances performance.

Reason To Accept:
1.A new dataset has been created for the multimodal fine-grained entity typing task, and the proposed model demonstrates a significant improvement in performance through the incorporation of object-level visual information. The MFIGER dataset holds promise as a valuable benchmark for future research and is a beneficial contribution to the community.
2.The motivation behind this work is clear. It marks the first attempt to introduce the multimodal fine-grained entity typing task, and the paper is well-written and easy to follow.
3.The reported results showcase substantial improvements: 11.27% improvement in total level, 1.91% in coarse level, and 10.46% in fine-grained level strict accuracy over baselines on MFIGER. Additionally, the attention visualization enhances the interpretability of the proposed model.

Reason To Reject:
1.The use of cross-modal attention layers for fusing object information is a common practice in multi-modal information extraction. It's advisable to reference relevant literature in the missing references section. Additionally, the hybrid classification layer appears to align with previous work in the field.
2.Several comparison experiments with baselines are lacking. 1)  For other text-only methods, it would be valuable to assess whether VinVL features can enhance text features through attention, ensuring a fairer comparison. 2) The experiments are conducted on a single dataset, and it's advisable to implement more multi-modal baselines for a comprehensive performance comparison.
3.Missing ablation study on the latent type classifier. It is necessary to investigate whether the inclusion of the latent type classifier contributes to an improvement in classification accuracy.

---

### Decision · Program_Chairs · 2023-10-07

**Decision:**

Accept-Findings

**Comment:**

Summary:
This paper begins by creating a new dataset for multimodal fine-grained entity typing, building upon FIGER. The authors introduce a multimodal object-level visual context network, which employs attention layers to fuse textual information and object-level visual data, facilitating the capture of fine-grained semantic information. Comparative experiments with several existing textual-level models demonstrate that the integration of object-level visual information enhances performance.

Reason To Accept:
1.A new dataset has been created for the multimodal fine-grained entity typing task, and the proposed model demonstrates a significant improvement in performance through the incorporation of object-level visual information. The MFIGER dataset holds promise as a valuable benchmark for future research and is a beneficial contribution to the community.
2.The motivation behind this work is clear. It marks the first attempt to introduce the multimodal fine-grained entity typing task, and the paper is well-written and easy to follow.
3.The reported results showcase substantial improvements: 11.27% improvement in total level, 1.91% in coarse level, and 10.46% in fine-grained level strict accuracy over baselines on MFIGER. Additionally, the attention visualization enhances the interpretability of the proposed model.

Reason To Reject:
1.The use of cross-modal attention layers for fusing object information is a common practice in multi-modal information extraction. It's advisable to reference relevant literature in the missing references section. Additionally, the hybrid classification layer appears to align with previous work in the field.
2.Several comparison experiments with baselines are lacking. 1)  For other text-only methods, it would be valuable to assess whether VinVL features can enhance text features through attention, ensuring a fairer comparison. 2) The experiments are conducted on a single dataset, and it's advisable to implement more multi-modal baselines for a comprehensive performance comparison.
3.Missing ablation study on the latent type classifier. It is necessary to investigate whether the inclusion of the latent type classifier contributes to an improvement in classification accuracy.